# Transcriptomic Insights into the Effects of Inoculation Density in *Areca catechu* Tissue Culture

**DOI:** 10.3390/plants14193073

**Published:** 2025-10-04

**Authors:** Jinqi Yan, Yu Li, Zijia Liu, Yusheng Zheng, Jixin Zou, Dongdong Li

**Affiliations:** 1National Key Laboratory for Tropical Crop Breeding, College of Tropical Agriculture and Forestry, Hainan University, Sanya 572025, China; yanjinqi_990804@163.com (J.Y.); liyutz2001@163.com (Y.L.); 183478@hainanu.edu.cn (Z.L.); yusheng.zheng@hainanu.edu.cn (Y.Z.); 2Coconut Research Institute of Chinese Academy of Tropical Agricultural Sciences (CATAS), Wenchang 571336, China

**Keywords:** *Areca catechu*, embryoid proliferation, plant hormone signaling, AcGIF1 transcription factor, RNA sequencing (RNA-seq)

## Abstract

Tissue culture technology represents a promising strategy for addressing the supply constraints of Areca catechu seedlings. Significant differences in embryoid proliferation were observed between isolated (CK) and aggregated (GL) culture conditions during subculture. To elucidate the underlying mechanisms, transcriptomic analysis was performed. Growth analysis indicated that GL embryoids exhibited the highest growth rate (50.2%) between days 12 and 15, with a peak proliferation efficiency of 50.52%. KEGG analysis identified plant hormone signaling as a key pathway. ELISA quantification demonstrated consistently higher JA levels in CK embryos, peaking at 294.06 ng/g on day 15, while IAA levels were significantly elevated in GL embryos (46.42 ng/g on day 15). The transcription factor AcGIF1 was identified as a central regulator, with further experiments confirming that JA negatively regulates and IAA positively regulates its expression. This study provides critical insights into the molecular mechanisms governing embryoid proliferation in response to inoculation density.

## 1. Introduction

*Areca catechu*, commonly known as the betel nut palm, is a significant agricultural crop in tropical regions, valued for its economic and cultural importance. The plant holds significant cultural and economic value in tropical regions of Asia, primarily due to the traditional use of its seeds as a stimulant and their incorporation in various herbal medicine practices [1,2]. The global betel nut industry has seen steady growth over the past decades, driven by increasing demand in both traditional and emerging markets. In countries such as India, Indonesia, and Bangladesh, betel nut production contributes significantly to rural economies, providing employment opportunities and income for smallholder farmers [3]. Nonetheless, conventional propagation methods are insufficient to meet demand, and production is severely threatened by diseases such as yellow leaf disease [4].

Given its widespread use and economic significance, there is a growing interest in improving the cultivation and propagation techniques of *Areca catechu* to meet increasing demand and ensure sustainable production [5]. In vitro tissue culture offers a promising strategy for the rapid mass production of uniform, disease-free planting material, which is crucial for both conservation efforts and sustainable cultivation [6]. These techniques not only facilitate the conservation of genetic resources but also enable the production of disease-free plants, which is particularly important given the susceptibility of *Areca catechu* to pests and diseases such as yellow leaf disease [7]. However, the efficiency of these methods is influenced by various factors, including the density of embryoid inoculation during the tissue culture process [8]. Understanding and optimizing these factors is critical for improving the success rate of tissue culture protocols.

The proliferation of callus tissue, a critical step in tissue culture, is influenced by multiple factors, including nutrient composition, growth regulators, and environmental conditions [9]. For instance, the balance of auxins and cytokinins in the culture medium has been shown to significantly affect callus growth and embryoid formation [10]. Additionally, light intensity, temperature, and pH levels can modulate the metabolic activity of cultured tissues, further influencing their proliferation and differentiation [11]. Understanding the molecular mechanisms underlying these factors is essential for optimizing tissue culture protocols. Transcriptomic analysis has emerged as a powerful tool for identifying differentially expressed genes and elucidating the metabolic pathways involved in callus proliferation and differentiation [12]. For example, studies on other plant species, such as Arabidopsis and rice, have revealed key genes involved in cell division, hormone signaling, and stress responses during tissue culture [13].

Preliminary observations in our lab revealed that embryoids inoculated in aggregates (high density) exhibited a significantly higher proliferation rate compared to those cultured in isolation (low density). This suggests that density-dependent cues, potentially involving the exchange of signaling molecules like phytohormones or juxtracrine communication, play a crucial role in regulating embryoid development. Optimizing this parameter could therefore lead to substantial improvements in propagation efficiency. However, the molecular mechanisms underlying this “aggregation effect” remain entirely unexplored in *A. catechu*. Therefore, the primary objective of this study is to investigate the molecular basis of the observed differences in embryoid differentiation rates between single and clustered inoculations. By conducting transcriptomic analysis at different stages of tissue culture, we aim to identify key genes and metabolic pathways associated with the differential responses to inoculation density. Specifically, we will explore the roles of hormone signaling pathways, cell cycle regulators, and stress-responsive genes in mediating the effects of inoculation density on embryoid differentiation. This research will provide valuable insights into the factors influencing the proliferation and differentiation of *Areca catechu* embryoids, contributing to the development of more efficient tissue culture protocols and enhancing the overall productivity of *Areca catechu* cultivation.

## 2. Results

### 2.1. Effect of Inoculation Density on Embryogenic Callus Proliferation

The phenotypic characteristics and proliferation efficiency of embryogenic callus in both the control (CK) and experimental (GL) groups were observed over five time points, revealing that while both groups exhibited increased volume and proliferation, the GL group showed more pronounced growth. Initial proliferation signs in the GL group appeared by day 12, with small, white granular globular embryos emerging, and their number and size continued to grow through days 15 and 18, though the proliferation rate slowed by day 21 (Figure 1). Fresh weight measurements indicated that the GL group consistently outweighed the CK group at each time point, with the most significant weight increase in the CK group occurring between days 9 and 12 (23.7%) and in the GL group between days 12 and 15 (50.2%) (Figure 2A). The average proliferation rates further confirmed the superior efficiency of the GL group, peaking at 50.52% on day 15, compared to the CK group’s highest rate of 26.60% (Figure 2B). These results demonstrate that high-density inoculation significantly enhances the growth and proliferation efficiency of *Areca catechu* embryogenic callus, with aggregated conditions yielding better outcomes than individual cultivation.

### 2.2. Analysis of Endogenous Hormone Content

The contents of jasmonic acid (JA), indole-3-acetic acid (IAA), cytokinin (CTK), and gibberellin (GA) in embryogenic callus were quantified using ELISA. JA levels in the control group (CK) peaked at day 15 (294.06 ng/g) and were consistently higher than in the treatment group (GL), which showed a declining trend from days 12 to 18, indicating significantly reduced JA levels under aggregated conditions (Figure 3A). Meanwhile, GA levels were consistently the lowest among the hormones measured and, crucially, showed no statistically significant differences between the CK and GL groups at any time point (Figure 3B). This suggests that GA dynamics are not a primary factor mediating the differential proliferation response to inoculation density in this specific experimental context (Figure 3B). In striking contrast, cytokinin (CTK) content was significantly and consistently higher in the GL group across multiple time points, reaching its peak at day 18 (9.85 ng/g) (Figure 3C). This pronounced difference aligns with the well-established role of CTK as a potent promoter of cell division and suggests its active involvement in the enhanced proliferation observed under aggregated conditions. Moreover, IAA levels in the CK group ranged from 14.98 ng/g to 22.15 ng/g, while the GL group exhibited higher levels, peaking at day 15 (46.42 ng/g), suggesting a positive correlation between IAA concentration and callus growth (Figure 3D). These results demonstrate dynamic changes in endogenous hormone levels and their roles in regulating embryogenic callus growth under different culture conditions.

### 2.3. Analysis of Differentially Expressed Genes (DEGs)

Differentially expressed genes (DEGs) were identified using thresholds of FDR < 0.01 and fold change (FC) ≥ 2 across five time points, comparing control (CK) and aggregated (GL) conditions. Significant enrichment of DEGs was observed under aggregated conditions (Figure 4A). Notably, CK12 vs. GL12 and CK15 vs. GL15 exhibited the highest number of DEGs, suggesting their critical role in regulating embryogenic callus development. A Venn diagram revealed 122 shared DEGs between CK12 vs. GL12 and CK15 vs. GL15, and 82 shared DEGs between CK15 vs. GL15 and CK18 vs. GL18, indicating that these periods involve a greater number of genes in the growth and development of *Areca catechu* embryogenic callus (Figure 4B).

### 2.4. KEGG Pathway Enrichment Analysis

KEGG pathway enrichment analysis identified significant metabolic pathways at 12, 15, and 18 days post-inoculation. At day 12 (CK12 vs. GL12), 18 pathways were enriched, including flavonoid biosynthesis (ko00941) and biosynthesis of secondary metabolites (ko01110), with the latter involving 39 genes (46.9%) and plant hormone signal transduction comprising 8 genes (9.6%) (Figure 5A). By day 15 (CK15 vs. GL15), pathways such as biosynthesis of secondary metabolites (95 genes, 46.5%), phenylpropanoid biosynthesis (19 genes, 9.3%), and plant hormone signal transduction (16 genes, 7.8%) were enriched (Figure 5B). At day 18 (CK18 vs. GL18), 17 pathways were enriched, including biosynthesis of secondary metabolites (28 genes, 51.8%) and phenylpropanoid biosynthesis (7 genes, 12.9%) (Figure 5C). Notably, the plant hormone signal transduction pathway (ko04075) was consistently enriched across all time points, underscoring its critical role in cell proliferation and plant growth, and highlighting it as a key focus for further research.

At day 12, IAA receptor genes were upregulated, while the TGA receptor gene TGA10 was downregulated, and TCH4 family genes (XTH22, XTH23, XTH1, and XTH7) were upregulated. At day 15, two JA receptor genes (TIFY10A) were downregulated, while IAA signaling components (IAA10, IAA20, GH3.17, ARF16, ARF9, and ARF15) were upregulated, and IAA25, AUX28, RR9, RR4, and RR10 were downregulated. The TCH4 genes XTH7 and XTH22 also showed upregulated expression. At day 18, the JA receptor gene TIFY10A was downregulated, while two IAA-related genes (GH3.8 and GH3.1) were upregulated. The downregulation of TIFY10A suggests that aggregated conditions suppress JA signaling, whereas the predominance of upregulated IAA receptor genes across all time points indicates enhanced IAA biosynthesis, supporting callus proliferation under aggregated conditions. These results underscore the differential regulation of hormone signaling pathways in response to aggregation, with IAA playing a key role in promoting growth.

### 2.5. Transcription Factor (TF) Analysis

Significant differences in transcription factors (TFs) were identified across five comparison groups (CK9 vs. GL9, CK12 vs. GL12, CK15 vs. GL15, CK18 vs. GL18, and CK21 vs. GL21), with 3, 29, 100, 20, and 31 differentially expressed TFs, respectively. The CK15 vs. GL15 group exhibited the most TFs, including 32 upregulated and 68 downregulated (Figure 6A). Transcriptome analysis and qRT-PCR validation highlighted that GIF1, GRF2, and bHLH94 showed significantly higher differential expression than other TFs, alongside DEGs such as IAA10, SAUR2, ERF5, TIFY4, and GH3. Pearson correlation analysis revealed GIF1 had a strong positive correlation with GH3 (*p* < 0.01) and a significant negative correlation with TIFY4 (*p* < 0.01), while showing negative correlations with IAA10, ERF5, and SAUR2 (*p* < 0.05). bHLH94 was positively correlated with IAA10 and SAUR2 (*p* < 0.05) but negatively correlated with TIFY4 (*p* < 0.01), GH3, and ERF5 (*p* < 0.05). GRF2 showed a positive correlation with ERF5 (*p* < 0.05) and significant negative correlations with TIFY4 and GH3 (*p* < 0.01), as well as negative correlations with IAA10 and SAUR2 (*p* < 0.05). Among the TFs, GIF1 exhibited the strongest correlations with DEGs, suggesting its potential as a key regulator of embryogenic callus proliferation differences (Figure 6B). The complete numerical data for the number of up- and down-regulated DEGs and TFs for each comparison are provided in Appendix A. These findings provide insights into the regulatory networks underlying *Areca catechu* embryogenic callus proliferation.

### 2.6. qRT-PCR Analysis

To explore the molecular mechanisms underlying the aggregation effect, eight differentially expressed genes (DEGs) were selected based on GO and KEGG analyses and validated using qRT-PCR (Figure 7). Among these, the transcription factor GIF1 showed significantly higher expression in the GL treatment group compared to the CK control group, with its expression pattern across five time points closely matching transcriptome data, suggesting its critical role in the aggregation effect. The transcription factor bHLH94 exhibited a unique expression trend, increasing initially before declining, indicating its potential regulatory role in different growth stages. The jasmonic acid-related factor TIFY10A displayed a negative correlation with transcriptome data, implying an inhibitory role of jasmonic acid signaling in the aggregation effect. In contrast, auxin-related factors (GH3.17, IAA10, SAUR2, GH3.8) and the cytokinin-related factor ARR9 showed positive correlations, suggesting that auxin and cytokinin signaling pathways may promote the aggregation effect by regulating cell growth and differentiation. Overall, the expression trends of the eight DEGs aligned well with transcriptome FPKM values, confirming the reliability of the data and providing a foundation for further investigation into the molecular mechanisms of the aggregation effect.

### 2.7. Effect of Jasmonic Acid on AcGIF1 Expression

To investigate the effect of jasmonic acid (JA) on AcGIF1 expression in *Areca catechu* embryogenic callus, different JA concentrations (10, 50, and 100 μmol/L) were added to the A06 subculture medium, with ultrapure water as the control. Under control conditions, AcGIF1 expression was highest in both CK and GL groups, peaking at 258 in the GL15 sample. With 10 μmol/L JA, AcGIF1 expression in GL15 decreased by 50.3% to 128; at 50 μmol/L JA, it dropped by 68.6% to 81; and at 100 μmol/L JA, it declined by 91.4% to 22. These results indicate that AcGIF1 expression decreases with increasing JA concentrations (Figure 8A). Notably, the GL15 sample consistently showed the highest AcGIF1 expression across all treatments, suggesting a critical role for AcGIF1 during this developmental stage and highlighting JA’s negative regulatory effect on its expression.

### 2.8. Effect of Indole-3-Acetic Acid (IAA) on AcGIF1 Expression

To explore the regulatory effects of indole-3-acetic acid (IAA) on AcGIF1 expression in *Areca catechu* embryogenic callus, varying IAA concentrations were added to the A06 subculture medium. At days 9, 12, and 21, AcGIF1 expression in the control (CK) and treatment (GL) groups showed no significant differences. However, at days 15 and 18, GL group expression diverged significantly, peaking at 29 on day 15, indicating strong regulatory effects of GL treatment at these stages (Figure 8B). Under 0.5 mg/L IAA, the CK group exhibited dynamic expression changes, while the GL group showed a gradient pattern, peaking at day 12 (216). At 1.0 mg/L IAA, AcGIF1 expression in the GL group reached 389 on day 15, highlighting a pronounced regulatory effect. Under 2.0 mg/L IAA, expression was significantly suppressed in both groups, with GL showing notable differences at days 12, 15, and 18. These findings demonstrate that IAA concentration and GL treatment differentially regulate AcGIF1 expression at specific time points, providing key insights into IAA’s role in *Areca catechu* embryogenic callus development.

**Figure 7 plants-14-03073-f007:**
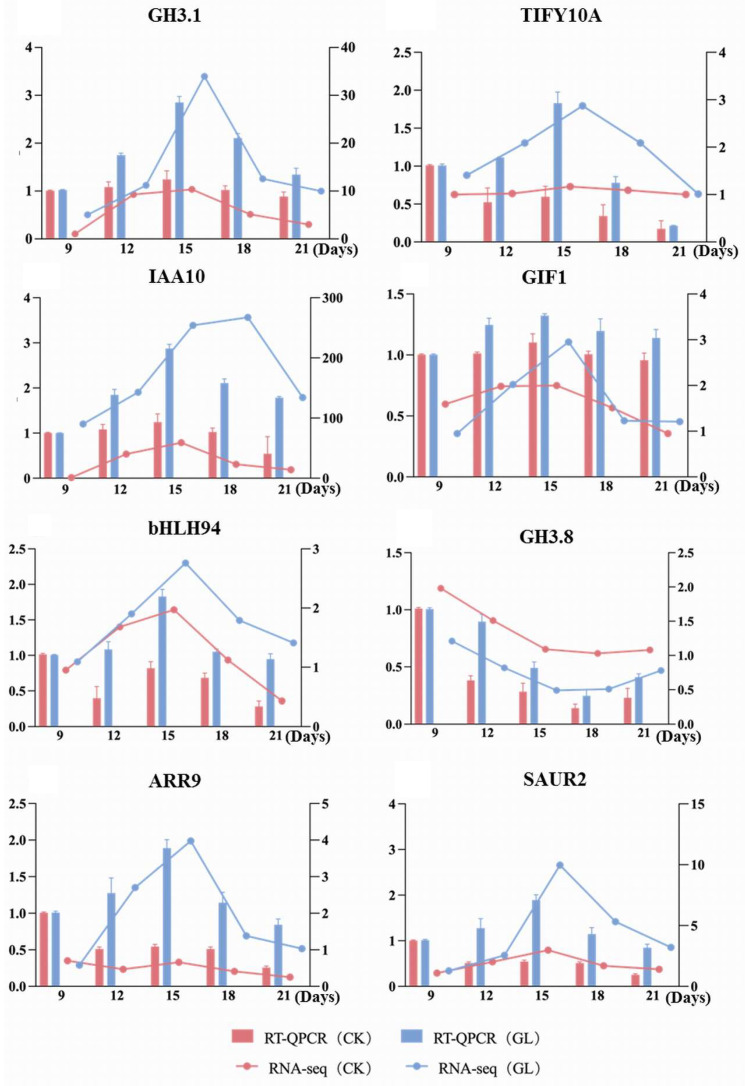
Verification of DEGs by qRT-PCR between CK and GL groups.

**Figure 8 plants-14-03073-f008:**
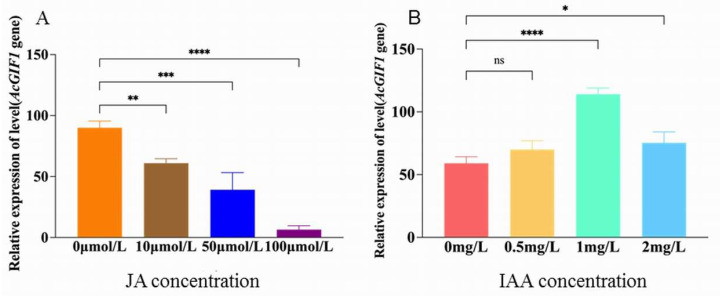
Expression level of AcGIF1 gene after exogenous hormone application. (**A**). Jasmonic acid; (**B**). IAA. * *p* < 0.05 (Significant); ** *p* < 0.01 (Highly Significant); *** *p* < 0.001 (Extremely Significant); **** *p* < 0.0005 (Extremely Highly Significant).

## 3. Discussion

This study revealed significant differences in the proliferation of *Areca catechu* embryogenic callus between the control (CK) and experimental (GL) groups during subculture, with the GL group showing the most pronounced weight increase (50.2%) between days 12 and 15 and the highest proliferation efficiency (50.52%) at day 15. Transcriptome sequencing identified 3662 differentially expressed genes (DEGs), with the most DEGs enriched at days 12, 15, and 18, particularly in the plant hormone signal transduction pathway. ELISA analysis confirmed that JA levels were downregulated and IAA levels were upregulated in the GL group, while CTK and GA showed no significant differences. Further analysis identified AcGIF1 as a key transcription factor involved in hormone signaling, with qRT-PCR validation showing that JA negatively regulates AcGIF1, whereas IAA positively correlates with its expression. These findings align with previous studies and provide insights into the regulatory roles of JA and IAA in *Areca catechu* embryogenic callus proliferation. Although the negative correlation between JA levels and AcGIF1 expression, coupled with its suppression by exogenous JA application, strongly suggests a negative regulatory relationship, the precise mechanistic causality warrants further investigation. Therefore, while our data robustly support JA and IAA as key regulators of AcGIF1, future studies employing techniques such as genetic approaches to manipulate AcGIF1 expression and hormone biosynthesis genes specifically in the embryoids, would be necessary to unequivocally establish direct causality.

The hormonal landscape influencing embryoid proliferation appears complex and involves more than just the JA-IAA interplay. Notably, cytokinin (CTK) content was significantly higher in the aggregated (GL) group. Given CTK’s well-established role in promoting cell division, its synergistic interaction with auxin signaling pathways is likely a significant contributor to the enhanced proliferation observed under high-density conditions. The convergence of elevated IAA and CTK signals could potently activate cell cycle regulators, a process potentially fine-tuned by AcGIF1. Conversely, gibberellin (GA) levels showed no significant differences between groups, suggesting it may play a less direct role in mediating the specific density effect reported here, though it remains crucial for other aspects of growth and development. This multi-hormone perspective suggests that the aggregation effect creates a unique hormonal niche—characterized by high IAA/CTK and low JA—that is highly conducive to proliferation.

Previous studies in model plants like Arabidopsis and various monocots have demonstrated that GIF1 regulates the development of tissues such as leaves, seeds, roots, and flowers [14]. As a transcriptional coactivator, GIF1 collaborates with GIF2 and GIF3 to control cell proliferation and organ size, with GIF1 being the most extensively studied due to its well-defined and broad regulatory roles. For example, in the cytokinin signaling pathway [15], PbRR1 activates PbGIF1 transcription, promoting pear fruit development and highlighting GIF1’s central role in developmental networks [16]. Additionally, GIF1 interacts with proteins involved in carbohydrate metabolism, protein turnover, cell division, and signaling, indicating its involvement in multiple physiological processes [17]. In this study, AcGIF1 exhibited high expression across all five developmental stages of *Areca catechu* embryogenic callus, suggesting it promotes cell proliferation by activating cell cycle-related genes, thereby regulating tissue and organ development in embryogenic callus.

Auxin plays a crucial role in plant development by regulating cell division, elongation, and differentiation through polar transport and metabolic homeostasis [18]. The GH3 gene family is essential in auxin metabolism, first identified in Glycine max and later found in both dicots and monocots [19]. In Arabidopsis, AtGH3.1 and AtGH3.6 participate in auxin signaling by catalyzing auxin conjugation with amino acids, reducing free auxin levels and mediating hormone crosstalk [20]. GH3 also negatively regulates lateral root formation and, along with PIN5, responds to auxin fluctuations to control meristem size [21]. In this study, qRT-PCR confirmed high GH3 expression during *Areca catechu* embryoid growth, while ELISA demonstrated its role in converting IAA into its inactive form (IAA-Glu), aligning with previous findings. These results highlight AcGH3 as a key regulator of auxin signaling and suggest its potential interaction with AcGIF1 in plant growth control. This discovery provides new insights into auxin-mediated regulatory networks and lays a foundation for further research into the complex mechanisms governing plant development.

Jasmonic acid (JA) plays a crucial role in plant growth and development, with JAZ proteins acting as key repressors in the JA signaling pathway [22]. When JA levels rise, JAZ proteins undergo ubiquitination and degradation, activating JA-responsive genes involved in growth regulation and defense. Studies show that COI1 mutants in Arabidopsis exhibit enhanced apical dominance, while JAZ mutants reduce branching under exogenous JA treatment, indicating JAZ’s role in growth suppression [23]. Similarly, in pear, PcCOI1 is highly expressed during branching, while PcJAZ remains low, suggesting a negative regulatory mechanism [24]. In this study, JA signaling influenced areca embryoid proliferation, with two key genes downregulated. JA levels across five developmental stages showed a significant negative correlation with embryoid growth. Further, exogenous JA application reduced GIF1 expression, leading to decreased proliferation. This suggests JA suppresses GIF1, thereby inhibiting areca embryoid growth. While the exact mechanisms remain unclear, this study fills a gap in understanding JA’s role in embryoid development, providing insights into hormone regulation and potential strategies for optimizing areca tissue culture and genetic improvement.

Plant growth and development are regulated by the interplay between jasmonic acid (JA) and auxin signaling pathways, with COI1, MYC2, and JAZ as key factors [23]. Auxin signaling activates the IAA-TIR-AUX/IAA-ARF pathway, promoting JA biosynthesis, while JA enhances auxin-related gene expression. JA influences growth by triggering JAZ degradation, which activates MYC2, suppressing PTL1 and PTL2 to regulate root development, while also enhancing MYB21/24, promoting floral organ formation [25]. Additionally, ARF6 and ARF8 modulate petal and carpel growth by regulating JA levels [26]. Transcriptome analysis suggests that AcJAZ, a JA-negative regulator, and AcGH3, an auxin-positive regulator, play roles in *Areca catechu* embryoid aggregation. The principles uncovered—that minimizing JA signaling while promoting an IAA/CTK-rich environment enhances proliferation—are highly likely to inform strategies for the micropropagation of other recalcitrant palm species and tropical crops with similar tissue culture challenges, moving beyond empirical optimization to a more predictive, mechanism-based approach.

In summary, the findings presented here offer insights that operate on two levels. Firstly, they provide a species-specific molecular framework for understanding somatic embryogenesis in A. catechu, identifying AcGIF1 as a key regulatory node and characterizing the distinct hormonal milieu (high IAA/CTK, low JA) that underlies efficient proliferation in aggregated cultures. This is a significant step forward for this particular crop. On a broader level, this study reinforces and refines general principles in plant tissue culture. It underscores that physical culture conditions (e.g., inoculation density) can profoundly influence physiological outcomes by altering endogenous hormone signaling networks. The antagonistic role of JA in proliferation and the synergistic role of IAA and CTK are likely generalizable phenomena across many species, as evidenced by studies in model plants. Thus, this work not only addresses a specific problem in A. catechu propagation but also contributes to the fundamental understanding of how cell communication and hormonal crosstalk regulate development in vitro.

## 4. Materials and Methods

### 4.1. Plant Materials

The embryogenic callus of *Areca catechu* used in this study was successfully induced by the Laboratory of Tropical Palm and Lipid Metabolism Regulation at Hainan University. The experiment utilized young, white globular embryos cultured at 26 °C with a subculture interval of 21 days. After collection, the embryogenic materials were immediately freeze-dried and stored at −80 °C for subsequent experiments.

### 4.2. Embryogenic Callus Inoculation

The embryogenic callus was divided into two groups: a control group (CK), where the callus was placed individually, and an experimental group (GL), where the callus was aggregated. Samples were collected at five time points: 9, 12, 15, 18, and 21 days post-inoculation. For each time point and each condition (CK and GL), three independent biological replicates were established. Each biological replicate originated from a separately initiated and maintained culture flask, containing callus material derived from a unique starting embryo. This design ensured that the replicates were statistically independent. The samples were labeled as CK9-1, CK9-2, CK9-3, GL9-1, GL9-2, GL9-3, etc., accordingly. The weight growth rate and proliferation rate of the callus were calculated as follows: Weight growth rate = (Fresh weight after inoculation − Fresh weight before inoculation)/Fresh weight before inoculation × 100%; Proliferation rate = (Fresh weight of proliferated callus − Fresh weight at inoculation)/Fresh weight at inoculation × 100%

### 4.3. RNA Extraction

Total RNA was extracted from the CK and GL groups using the Fast Pure Plant Total RNA Isolation Kit (Polysaccharides & Polyphenolics-Rich) (Vazyme, Nanjing, China) following the manufacturer’s instructions. Briefly, the samples were flash-frozen in liquid nitrogen, ground into a fine powder, and processed for RNA extraction. The extracted RNA was stored at −80 °C, and its concentration and integrity were assessed prior to further analysis [8].

### 4.4. Transcriptome Sequencing and Data Analysis

Transcriptome sequencing was performed on all fifteen samples per condition (3 biological replicates × 5 time points). Each of the independent biological replicates described in Section 4.2 was processed and sequenced individually. This design allows for the statistical assessment of differential expression within and between groups. RNA sequencing (RNA-seq) was conducted by BGI Genomics (Shenzhen, China). Clean reads were aligned to the *Areca catechu* reference genome using HISAT2 software (2.2.1) to determine their genomic locations. The assembled genes were annotated and functionally analyzed using BLAST (2.17) against the *Areca catechu* database. To explore the biological functions and metabolic pathways of the genes, bioinformatics tools and databases were employed. Key biological processes and pathways were identified by selecting the top 20 enriched KEGG pathways with a significance threshold of *p* < 0.05. This analysis provided insights into the dynamic changes in biological functions and regulatory mechanisms during different developmental stages of *Areca catechu*.

### 4.5. Identification and Analysis of Differentially Expressed Genes (DEGs)

Differentially expressed genes (DEGs) were identified using the DESeq2 (vX.Y.Z) package in R (3.22). Genes with an adjusted *p*-value < 0.05 and an absolute log2 fold change (|log2FC|) ≥ 1 (equivalent to a linear fold change of ≥2) were considered statistically significant. DEGs were analyzed between the control and experimental groups at each time point (CK9 vs. GL9, CK12 vs. GL12, CK15 vs. GL15, CK18 vs. GL18, and CK21 vs. GL21). Hierarchical clustering was performed to visualize the expression patterns of DEGs. Additionally, Gene Ontology (GO) enrichment analysis and KEGG pathway analysis were conducted using a hypergeometric distribution model to elucidate the functional roles and metabolic pathways associated with the DEGs.

### 4.6. Measurement of Endogenous Hormone Levels

Approximately 1 g of sample was ground into a fine powder in liquid nitrogen. The homogenate was prepared using PBS (pH 7.2–7.4, 0.01 mol/L) at a 10% concentration, followed by centrifugation at 5000 rpm for 15 min at 4 °C. The supernatant was collected for analysis. Endogenous hormone levels were quantified using a double-antibody sandwich enzyme-linked immunosorbent assay (ELISA) kit (Jiangsu Enzyme Immunity Biotechnology Co., Ltd., Nanjing, China). For each treatment group and time point, measurements were performed on the three independent biological replicates.

### 4.7. Quantitative Real-Time PCR (qRT-PCR)

Primers were designed using Primer 5 software, with Ac-β-actin as the reference gene. qRT-PCR was performed using the 2xQ3 SYBR qPCR Master Mix kit. For each treatment group and time point, measurements were performed on the three independent biological replicates. The primers used for qRT-PCR are listed in Table 1.

### 4.8. Exogenous JA and IAA Treatment Experiment

To investigate the effects of jasmonic acid (JA) and indole-3-acetic acid (IAA) on AcGIF1 expression in betel nut embryoids, different concentrations of JA (10 μmol/L, 50 μmol/L, and 100 μmol/L) and IAA (0.5 mg/L, 1 mg/L, and 2 mg/L) were added to the A06 subculture medium, respectively. An equal volume of ddH_2_O was used as the control, with three biological replicates per treatment to ensure scientific reliability and reproducibility of the results.

### 4.9. Data Analysis

Data were statistically analyzed using Excel and IBM SPSS 20.0 software. Significant differences were determined at *p* < 0.05, *p* < 0.01, and *p* < 0.001. Pearson correlation coefficients were used to assess the relationship between transcription factors and hormone-related DEGs. Heatmaps illustrating the expression patterns of differentially expressed transcription factors were generated using TBtools (2.0). Figures were prepared using GraphPad Prism 9.

## Figures and Tables

**Figure 1 plants-14-03073-f001:**
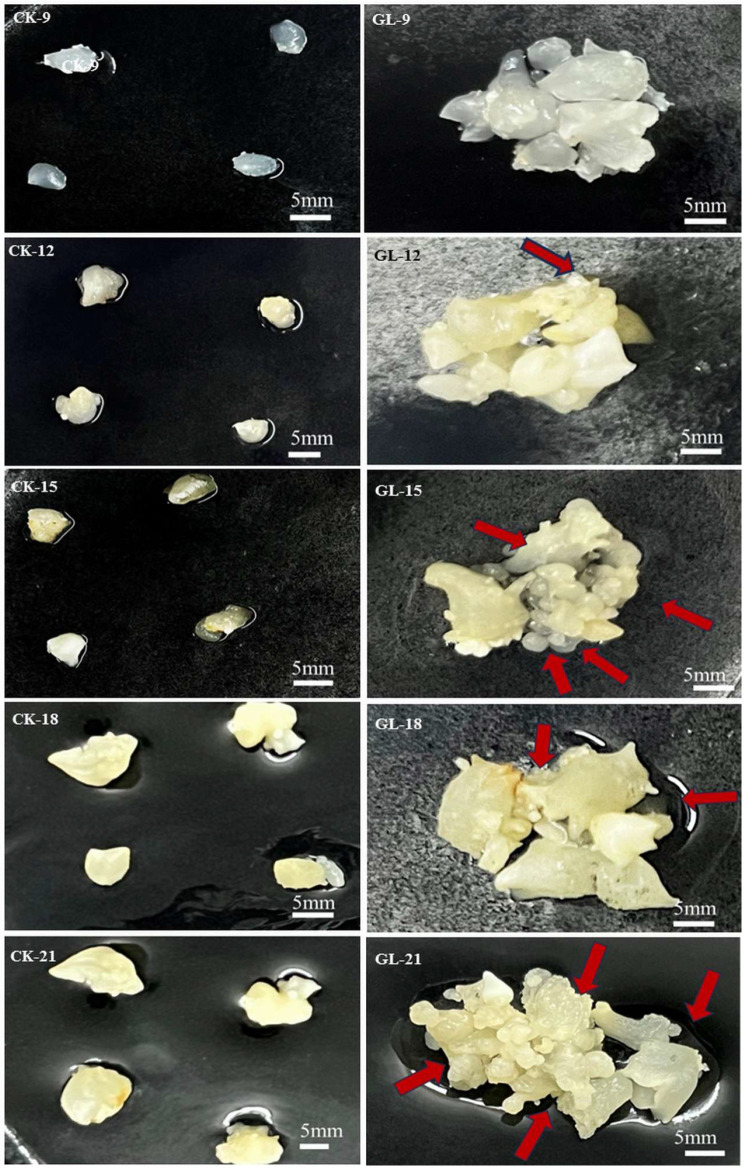
Effect of inoculation density on embryogenic callus proliferation. CK. individually (Control); GL. aggregated (experimental group). Red arrow represents newly formed callus.

**Figure 2 plants-14-03073-f002:**
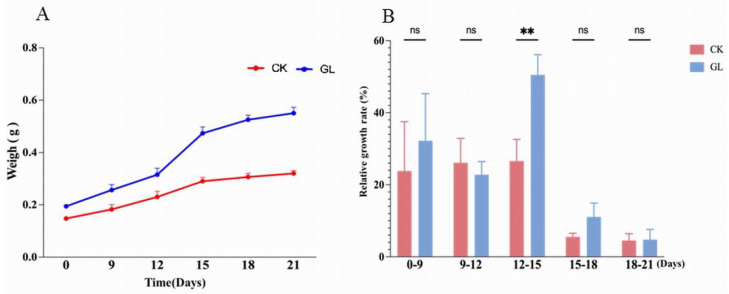
Changes of Areca nut embryoids under two different inoculation densities. (**A**). Weight changes; (**B**). Average proliferation rates. ** represents extremely significant difference; “ns” represents no significant difference.

**Figure 3 plants-14-03073-f003:**
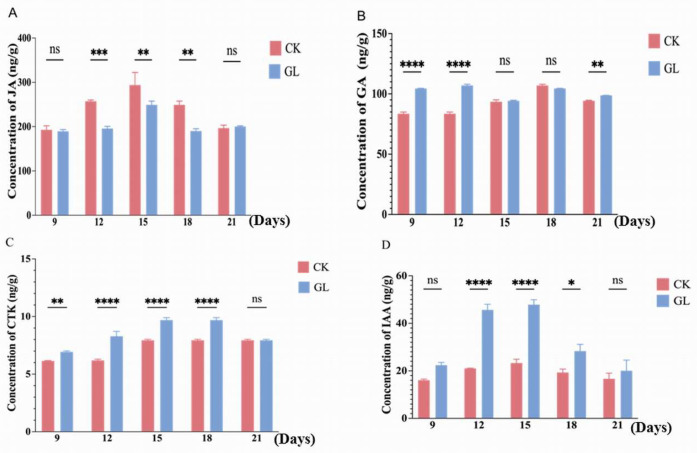
Hormone content in of Areca nut embryoids at two different inoculation densities. (**A**). Jasmonic acid; (**B**). Gibberellin; (**C**). Cytokinin; (**D**). IAA. * *p* < 0.05 (Significant); ** *p* < 0.01 (Highly Significant); *** *p* < 0.001 (Extremely Significant); **** *p* < 0.0005 (Extremely Highly Significant).

**Figure 4 plants-14-03073-f004:**
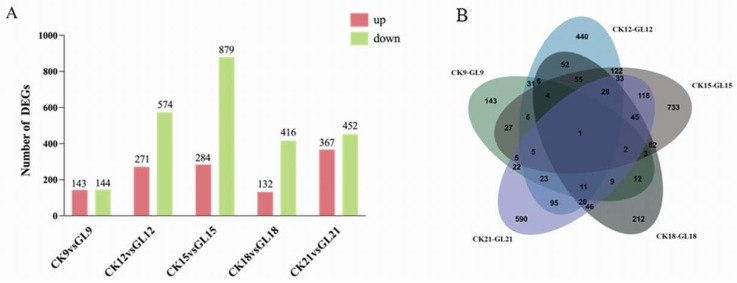
Differential genes in Areca nut embryoids under two different inoculation densities. (**A**). The number of up-regulated (light grey) and down-regulated (dark grey) differentially expressed genes (DEGs) for each comparison; (**B**). Compare the Venn diagram of 5 groups of differentially expressed genes.

**Figure 5 plants-14-03073-f005:**
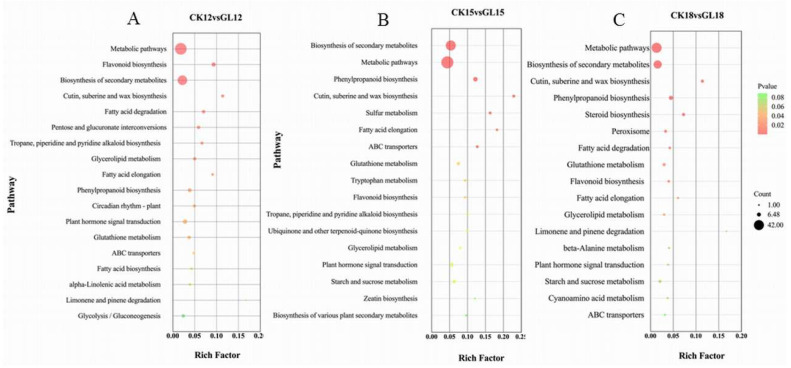
Results of KEGG Enrichment of Differential genes. (**A**). 12 days; (**B**). 15 days; (**C**). 18 days.

**Figure 6 plants-14-03073-f006:**
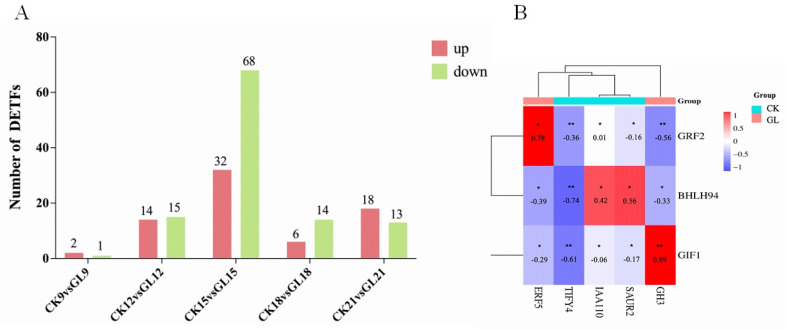
Differentially expressed transcription factors between CK and GL group. (**A**). Number of TFs. (**B**). Correlation coefficient diagram between TFs and DEGs. * *p* < 0.05 (Significant); ** *p* < 0.01 (Highly Significant).

**Table 1 plants-14-03073-t001:** Primers for gene qRT-PCR analysis.

Primere	Sequence (5′-3′)
Ac-Actin-F	TTCCAGCCTTCGCTCATT
Ac-Actin-R	CCTCCACCACTAAGCACAATG
bHLH94-F	GTAGGCGGAGCCATTGATTT
bHLH94-R	GCTGCTGGAGTGTCCTCTTCT
GIF1-F	GGCCCTCGGTATATGCAACA
GIF1-R	CAGCAAGCAGCACTGCATAG
IAA10-F	TGCCGTTGGGTCCACTAATC
IAA10-R	CCTCAAATGCTGCAGTCACG
TIFY10A-F	TCAGCAACTGCAGCAAGAGT
TIFY10A-R	CGACAGCCAACCTTTGATGC
GH3.8-F	GCTCATGGACTACGCCATCT
GH3.8-R	GTCGTCTCCACCCACTTCAG
SAUR2-F	CATATCGGTGTGCTTGCTGC
SAUR2-R	GGCAAGTACCACAAAGCTGC
ARR9-F	ACCGTGTCTTAGCTGTGGATG
ARR9-R	GCAGTTCCTGGGTCTACAGG
GH3.17-F	GTGATGAGCCAATTCGTGCC
GH3.17-R	ACTACAAGAGCCGCCACTTC

## Data Availability

Data will be made available on request.

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
