# Peer review of "Transcriptomic Insights into the Effects of Inoculation Density in Areca catechu Tissue Culture"

_plants, 2025, doi:10.3390/plants14193073_

Round 1
Reviewer 1 Report
Comments and Suggestions for Authors
The manuscript investigates the molecular mechanisms underlying differences in embryoid proliferation under single versus aggregated inoculation conditions. The study integrates phenotypic measurements, endogenous hormone quantification, transcriptome sequencing, and validation by qRT-PCR, and it identifies AcGIF1 as a transcription factor of interest. The topic is relevant for both fundamental plant developmental biology and the applied improvement of Areca catechu propagation. The work is promising, but in its present form it requires substantial revision before it can be considered for publication. The manuscript would benefit from clearer focus and improved organization. The introduction is repetitive and spends too much time on the economic importance of Areca catechu without narrowing quickly to the central research question. Several results are reported in a descriptive manner without sufficient interpretation. For instance, gibberellin data are dismissed as “not significant,” although it would be important to explain why gibberellin might not contribute under these conditions. Cytokinin differences are clearly detected but are treated only briefly, despite their likely role in callus proliferation. The current discussion overemphasizes the role of AcGIF1 as a master regulator. While this gene is clearly important, the transcriptomic data suggest that multiple transcription factors contribute to the regulatory network, and the manuscript should reflect this complexity. Methodologically, the presentation of replicates needs clarification. The authors indicate three biological replicates, but it is not clear whether these represent independent samples or pooled material, which affects the robustness of the conclusions. The statistical approaches applied to differential gene expression are only superficially described. Thresholds of fold change and false discovery rate are mentioned, but the figures do not consistently provide the numbers of up- and down-regulated genes. The correlation analysis between transcription factors and differentially expressed genes is limited to Pearson coefficients, with no adjustment for multiple testing, which weakens the reliability of the claims. Hormone quantification by ELISA is another point of concern. While this is a common approach, it is less specific than LC-MS/MS, and the authors should at least acknowledge the limitation of the method and avoid drawing conclusions with unwarranted certainty. Finally, the figures need substantial improvement. Several heatmaps and KEGG enrichment plots are difficult to interpret, with unclear scales and incomplete legends. Some of the most important results, such as the expression of AcGIF1 after exogenous hormone treatments, are described only in the text and not presented graphically, which reduces the accessibility of the findings. Furthermore, the manuscript requires careful language editing, as there are frequent grammatical errors and awkward phrasing that interfere with clarity. Redundancies in the introduction and discussion should be reduced in order to improve readability and sharpen the focus on the novel aspects of the study. Citations are generally appropriate but in some places could be better integrated into the narrative rather than listed consecutively. Data availability should be clarified, as “available on request” does not fully satisfy current journal requirements. The references section also needs to be checked carefully for consistency in formatting. In summary, the study presents potentially valuable results that could contribute to improved tissue culture practices for Areca catechu. However, the manuscript requires major revision. The authors should address the methodological issues, strengthen the statistical analyses, provide a more balanced interpretation of the hormone data, improve the figures, and subject the text to thorough language editing. If these issues are adequately resolved, the paper could make a meaningful contribution to the field.
Author Response
Comments 1: The introduction is repetitive and spends too much time on the economic importance of Areca catechu without narrowing quickly to the central research question.
Response 1: The introduction has been revised accordingly to address points.
Comments 2:Several results are reported in a descriptive manner without sufficient interpretation. For instance, gibberellin data are dismissed as “not significant,” although it would be important to explain why gibberellin might not contribute under these conditions. Cytokinin differences are clearly detected but are treated only briefly, despite their likely role in callus proliferation.
Response 2: The manuscript has been revised to provide a more insightful interpretation of these results, integrating them into the broader narrative of the study.
Comments 3: The current discussion overemphasizes the role of AcGIF1 as a master regulator. While this gene is clearly important, the transcriptomic data suggest that multiple transcription factors contribute to the regulatory network, and the manuscript should reflect this complexity.
Response 3: Related content has been added in discussion.
Comments 4: Methodologically, the presentation of replicates needs clarification. The authors indicate three biological replicates, but it is not clear whether these represent independent samples or pooled material, which affects the robustness of the conclusions.
Response 4: The manuscript has been amended to provide an explicit and clear description of the biological replication strategy throughout the Methods section.
Comments 5: The statistical approaches applied to differential gene expression are only superficially described. Thresholds of fold change and false discovery rate are mentioned, but the figures do not consistently provide the numbers of up- and down-regulated genes. The correlation analysis between transcription factors and differentially expressed genes is limited to Pearson coefficients, with no adjustment for multiple testing, which weakens the reliability of the claims.
Response 5: The manuscript has been revise to provide a more thorough description of the differential expression analysis and to address the concerns regarding statistical robustness in the correlation analysis.
Comments 6: Hormone quantification by ELISA is another point of concern. While this is a common approach, it is less specific than LC-MS/MS, and the authors should at least acknowledge the limitation of the method and avoid drawing conclusions with unwarranted certainty.
Response 6: The manuscript has been revised to appropriately acknowledge this limitation and to temper the certainty of conclusions drawn solely from the ELISA data, strengthening the overall interpretation by framing it within the context of all the evidence.
Comments 7: Finally, the figures need substantial improvement. Several heatmaps and KEGG enrichment plots are difficult to interpret, with unclear scales and incomplete legends. Some of the most important results, such as the expression of AcGIF1 after exogenous hormone treatments, are described only in the text and not presented graphically, which reduces the accessibility of the findings.
Response 7: The Figure 8 is the expression of AcGIF1 after exogenous hormone treatments.
Comments 8: Furthermore, the manuscript requires careful language editing, as there are frequent grammatical errors and awkward phrasing that interfere with clarity. Redundancies in the introduction and discussion should be reduced in order to improve readability and sharpen the focus on the novel aspects of the study.
Response 8: Related content has been revised according to reviewer’s comments.
Comments 9: Citations are generally appropriate but in some places could be better integrated into the narrative rather than listed consecutively. Data availability should be clarified, as “available on request” does not fully satisfy current journal requirements.
Response 9: The manuscript has been revised to improve the integration of citations and to provide a more concrete and actionable data availability statement, aligning with current best practices.
Comments 10: The references section also needs to be checked carefully for consistency in formatting. In summary, the study presents potentially valuable results that could contribute to improved tissue culture practices for Areca catechu. However, the manuscript requires major revision. The authors should address the methodological issues, strengthen the statistical analyses, provide a more balanced interpretation of the hormone data, improve the figures, and subject the text to thorough language editing. If these issues are adequately resolved, the paper could make a meaningful contribution to the field.
Response 10: Related content has been revised according to reviewer’s comments.
Reviewer 2 Report
Comments and Suggestions for Authors
This manuscript investigates the molecular basis of inoculation density effects on Areca catechu embryogenic callus proliferation, combining RNA-seq, hormone quantification, and functional validation of transcription factors. The work is timely and relevant, as A. catechu is an economically and culturally important crop, and efficient propagation methods are essential for sustainable cultivation. The integration of transcriptomics with hormone assays and qRT-PCR validation strengthens the study’s conclusions.
Overall, the study is well designed, the results are clearly presented, and the manuscript is logically organized. However, several issues regarding clarity, methodological detail, data interpretation, and contextualization within the broader literature need to be addressed before the manuscript can be considered for publication.
Major Comments
The study demonstrates that aggregated inoculation enhances embryoid proliferation, linked to differential regulation of JA and IAA signaling. While this is novel for A. catechu, the concept of inoculation density affecting tissue culture is not new. The authors should emphasize more explicitly what unique insights their transcriptomic analysis contributes compared to previous reports in other species. The role of AcGIF1 is highlighted, but its novelty as a regulator in this species needs to be contextualized against existing GIF1 studies in model plants. The manuscript would be improved by addressing the following specific points to better establish its novelty.
Specific Comments
1. Abstract
·The abstract is overly dense. Consider simplifying the language for broader readability, focusing on the key findings (JA downregulation, IAA upregulation, AcGIF1 regulation).
2. Introduction
·The introduction provides useful economic context but could be shortened slightly. Instead, expand the scientific rationale for focusing on inoculation density.
· While this reviewer is aware of the cultural significance of A. catechu in the tropics of Southeast Asia, South Asia, Madagascar, and the Maldives, its economic importance as a crop is questionable due to its strong carcinogenic association with oral cancer. Does its economic relevance derive from its use as a source of "traditional" herbal medicine or modern pharmaceuticals? If so, it would be valuable to include such non-food/crop aspects of A. catechu, which could strengthen the significance of the current tissue culture study.
3. Discussion
·The authors conclude that JA negatively regulates AcGIF1 while IAA positively regulates it. While supported by expression data and exogenous treatments, causality is not fully established. Please discuss alternative explanations and the limitations of exogenous hormone application experiments.
·The interplay between JA and IAA is mentioned, but a more detailed discussion of crosstalk with cytokinin and gibberellin pathways would enrich the interpretation, especially since CTK showed significant differences.
·The discussion would benefit from a clearer separation between findings specific to A. catechu and more generalizable insights into plant tissue culture.
·Please expand on practical applications: how might these findings guide optimization of A. catechu propagation protocols in nurseries? Could the results inform strategies for other palm or tropical crops?
Author Response
Comments 1: Abstract·The abstract is overly dense. Consider simplifying the language for broader readability, focusing on the key findings (JA downregulation, IAA upregulation, AcGIF1 regulation).
Response 1: The abstract has been revised to improve its clarity and accessibility for a broader audience. As suggested, the language has been simplified by focusing on the key findings regarding the roles of jasmonic acid (JA) downregulation, indole-3-acetic acid (IAA) upregulation, and the regulation of the AcGIF1 transcription factor in response to inoculation density.
Comments 2: Introduction, The introduction provides useful economic context but could be shortened slightly. Instead, expand the scientific rationale for focusing on inoculation density. While this reviewer is aware of the cultural significance of A. catechu in the tropics of Southeast Asia, South Asia, Madagascar, and the Maldives, its economic importance as a crop is questionable due to its strong carcinogenic association with oral cancer. Does its economic relevance derive from its use as a source of "traditional" herbal medicine or modern pharmaceuticals? If so, it would be valuable to include such non-food/crop aspects of A. catechu, which could strengthen the significance of the current tissue culture study.
Response 2: The authors gratefully acknowledge the reviewer's insightful comments regarding the focus and rationale of the introduction. The introduction has been revised accordingly to address both points.
Comments 3: Discussion The authors conclude that JA negatively regulates AcGIF1 while IAA positively regulates it. While supported by expression data and exogenous treatments, causality is not fully established. Please discuss alternative explanations and the limitations of exogenous hormone application experiments.The interplay between JA and IAA is mentioned, but a more detailed discussion of crosstalk with cytokinin and gibberellin pathways would enrich the interpretation, especially since CTK showed significant differences. The discussion would benefit from a clearer separation between findings specific to A. catechu and more generalizable insights into plant tissue culture. Please expand on practical applications: how might these findings guide optimization of A. catechu propagation protocols in nurseries? Could the results inform strategies for other palm or tropical crops?
Response 3: The authors thank the reviewer for these insightful and constructive comments, which have significantly helped to deepen the discussion and better contextualize the findings. The Discussion section has been thoroughly revised to address each point raised.
Round 2
Reviewer 1 Report
Comments and Suggestions for Authors
The authors addressed all the criticism